# Putative Transcription Factor Genes Associated with Regulation of Carotenoid Biosynthesis in Chili Pepper Fruits Revealed by RNA-Seq Coexpression Analysis

**DOI:** 10.3390/ijms231911774

**Published:** 2022-10-04

**Authors:** Maria Guadalupe Villa-Rivera, Octavio Martínez, Neftalí Ochoa-Alejo

**Affiliations:** 1Departamento de Ingeniería Genética, Unidad Irapuato, Centro de Investigación y de Estudios Avanzados del Instituto Politécnico Nacional, Irapuato 36824, Mexico; 2Unidad de Genómica Avanzada, Unidad Irapuato, Centro de Investigación y de Estudios Avanzados del Instituto Politécnico Nacional, Irapuato 36824, Mexico

**Keywords:** chili pepper, carotenoids, transcriptional regulation, RNA-Seq, transcription factors

## Abstract

During the ripening process, the pericarp of chili pepper (*Capsicum* spp.) fruits accumulates large amounts of carotenoids. Although the carotenoid biosynthesis pathway in the *Capsicum* genus has been widely studied from different perspectives, the transcriptional regulation of genes encoding carotenoid biosynthetic enzymes has not been elucidated in this fruit. We analyzed RNA-Seq transcriptomic data from the fruits of 12 accessions of *Capsicum annuum* during the growth, development, and ripening processes using the R package named *Salsa*. We performed coexpression analyses between the standardized expression of genes encoding carotenoid biosynthetic enzymes (target genes (TGs)) and the genes of all expressed transcription factors (TFs). Additionally, we analyzed the promoter region of each biosynthetic gene to identify putative binding sequences for each selected TF candidate. We selected 83 TFs as putative regulators of the carotenogenic structural genes. From them, putative binding sites in the promoters of the carotenoid-biosynthesis-related structural genes were found for only 54 TFs. These results could guide the search for transcription factors involved in the regulation of the carotenogenic pathway in chili pepper fruits and might facilitate the collection of corresponding experimental evidence to corroborate their participation in the regulation of this biosynthetic pathway in *Capsicum* spp.

## 1. Introduction

Chili pepper (*Capsicum* spp.) is a globally cultivated horticultural crop. Chili pepper fruits are popular because of their great variety of forms, sizes, flavors, and aromas, especially for their unique and characteristic pungency. Chili pepper pods produce a series of compounds, such as anthocyanins, carotenoids, vitamin C, flavonoids, which are powerful antioxidants, and hot capsaicinoid analogs, among others. Chili pepper represents an important model for studying the carotenogenesis process as well as investigating and establishing the carotenoid biosynthetic pathway (Figure 1) [1]. Carotenoids are the second most abundant pigments in nature behind chlorophylls. These pigments are responsible for the yellow, orange, and red colors of flowers and fruits and have important functions as attractors of pollinators, complementary pigments during photosynthesis, and efficient protectors of photosystems against the damage caused by the free radicals generated during the photosynthetic process. Nutraceutic benefits of chili pepper carotenoids for human health have been reported recently [2]. Although the carotenoid biosynthetic pathway in chili pepper fruits has been fully described, the regulation mechanisms at the transcriptional level of all carotenogenic genes are currently poorly understood [3]. Regarding this aspect, a tomato fruit gene regulatory network was generated by Pan et al. (2013) using artificial neural network inference analysis and transcription gene expression patterns derived from fruits sampled at various points during development and ripening, and one of the transcription factor gene expression profiles, with a sequence related to the Arabidopsis (*Arabidopsis thaliana*) *ARABIDOPSIS PSEUDO RESPONSE REGULATOR2-LIKE* gene (*APRR2-Like*), was upregulated at the breaker stage in wild-type tomato fruits and, when overexpressed in transgenic lines, increased the plastid number, area, and pigment content, enhancing the levels of chlorophyll in immature unripe fruits and carotenoids in red ripe fruits. Additionally, a putative ortholog of the tomato *APPR2-Like* gene in sweet pepper (*Capsicum annuum*) was associated with pigment accumulation in fruit tissues [4]. Song et al. (2020) reported an analysis of the *ERF* transcription factor gene family in the *Capsicum* genome to identify transcription factor genes possibly involved in the regulation of species-specific metabolites, including carotenoids, and they observed that the expression patterns of *CaERF82*, *CaERF97*, *CaERF66*, *CaERF107,* and *CaERF101* were consistent with the expression profiles of carotenoid-biosynthesis-related pathway genes and the accumulation of carotenoids (β-carotene, zeaxanthin, and capsorubin) in the pericarp of chili pepper fruits [5]. In a genome-wide identification and analysis of the *MYB* transcription factor gene family in chili pepper (*Capsicum* spp.), Arce-Rodríguez et al. (2021) sought to identify *CaMYB* genes that were positively correlated with the expression patterns of the carotenoid-related genes *PSY (phytoene synthase)*, *β-carotene hydroxylase 1* (*BCH1*), and *capsanthin-capsorubin synthase* (*CCS*) during the development and ripening of Serrano ‘Tampiqueño 74’ chili pepper fruits (0, 10, 20, 30, 40, 50, and 60 days postanthesis) as an approach to predict transcription factors possibly involved in the regulation of this pathway, and they found that the transcriptomic expression patterns of those genes clustered with the *MYB*-related genes *CaDIV1*, *CaDIV3*, *CaMYBR13*, *CaTRF2*, *CaMYBC1*, and *CaPHR9* and an atypical *MYB* (*CaMYB5R*) [6]. Li et al. (2021) carried out a transcriptome analysis and carotenoid content determination in orange and red chili pepper fruits at three developmental stages (25, 40, and 55 days after flowering) and found that the expression patterns of the *BCH1* and *CCS* genes were significantly different during fruit development. Moreover, through a weighted gene coexpression network analysis (WGCNA), the transcription factor genes F-box protein *SKIP23, GATA transcription factor 26, U-box domain-containing protein 52*, zinc finger family *FYVE/PHD-type, RING/FYVE/PHD-type,* and *CONSTANS-LIKE 9* were proposed to be involved in the regulation of the *CCS* gene [7]. More recently, Ma et al. (2022) analyzed the expression patterns of the transcription factor gene *CaMYB306* in chili pepper fruits of contrasting colors (‘XHB’ (red fruits) and mutant H0809 pepper (yellow fruits)) at three developmental stages (25, 40, and 55 days after flowering). *CaMYB306* expression was significantly higher in ‘XHB’ fruit than in mutant H0809 fruit. A knockdown of *CaMYB306* by gene silencing inhibited carotenoid biosynthesis and markedly decreased the expression of carotenoid-related genes (*CaPSY*, *CaBCH,* and *CaZEP*) in chili pepper fruit. Furthermore, *CaMYB306*-overexpressing (*CaMYB306*-OE) ‘Micro-Tom’ tomato plants showed earlier coloration with increased carotenoid accumulation. The expression levels of key carotenoid-biosynthesis-related genes (*SlPSY* and *SlPDS*) were increased in *CaMYB306*-OE fruit [8].

However, in addition to these findings in chili pepper fruits, transcriptional regulation of carotenoid biosynthesis seems to be a very complex process since diverse transcription factors have been proposed as regulators of this pathway in different plant species using different experimental approaches; for example, Welsch et al. (2007) demonstrated that AtRAP2.2, a member of the APETALA2 (AP2)/ethylene-responsive element-binding protein transcription family, showed positive binding to the ATCTA cis-element present in the promoter of the *phytoene desaturase* (*PDS*) gene in *Arabidopsis* [9]. Similar interactions between transcription factors and promoters of carotenoid biosynthetic genes have been reported as the mechanism regulating this pathway, and the tomato MADS-box transcription factor RIPENING INHIBITOR (RIN) was found to interact with the promoter of the *PSY1* gene, which was enriched following a chromatin immunoprecipitation assay (ChIP); moreover, the *PSY1* gene showed an increase in expression at the onset of ripening in wild-type but not in *rin* mutant fruits [10]. According to Mohanty et al. (2016), the promoter analysis of genes upregulated in response to blue and red lights in rice leaves revealed the involvement of key transcription factors, such as bHLH, bZIP, MYB, WRKY, ZnF, and ERF (jasmonic-acid-inducible), in the regulation of a higher accumulation of carotenoids and phenolic compounds under blue light [11]. In papaya (*Carica papaya*), Fu et al. (2017) indicated that CpEIN3a regulated the carotenoid-biosynthesis-related genes *CpPDS4* and *CpBCH* during fruit ripening, as demonstrated by electrophoretic mobility shift assays (EMSA) and transient expression analyses. Moreover, CpEIN3a physically interacted with the transcription factor CpNAC2, which acted as a transcriptional activator of *CpPDS2/4*, *CpZDS*, *CpLCYE,* and *CpBCH* by directly binding to their promoters [12]. According to Lu et al. (2018), the promoter sequences of the *lycopene β-cyclase* genes *LCYB1* and *LCYB2* were used as bait in a yeast one-hybrid screen cDNA library for promoter-binding proteins from sweet orange (*Citrus sinensis*) to identify TFs that might regulate carotenoid metabolism, and they isolated the transcription factor gene *CsMADS6*, which was demonstrated to modulate carotenoid metabolism since its overexpression increased the expression of the *PSY*, *PDS*, *carotenoid cleavage*
*dioxygenase 1* (CCD1), and *LCYB1* genes in transformed overexpressing transgenic citrus calli and tomato (*Solanum lycopersicum*) plants. Furthermore, CsMADS6 upregulated the expression of the *PSY*, *PDS,* and *CCD1* genes by directly binding to their promoters [13]. Ampomah-Dwamena et al. (2019) identified and characterized the kiwifruit TF *MYB7* gene, whose protein product was found to activate the promoter of the kiwifruit *AdLCYb* gene [14]. Recently, Y. Liu et al. (2021) demonstrated the participation of the SlGRAS4 transcription factor as a regulator of carotenoid biosynthesis by overexpressing this gene in tomato. The fruits from overexpressing tomato plants showed an increase in carotenoids and a higher expression of *PSY1* than those from nontransformed plants, and the expression of the negative regulator of tomato fruit ripening, *SlMADS1*, was repressed [15]. Another approach to search for transcription factors possibly involved in the transcriptional regulation of carotenoid-biosynthesis-related genes has been the comparative expression pattern analysis of structural biosynthetic genes and transcription factor genes in color-contrasting fruits or organs and by coexpression analyses during fruit development and ripening; for example, Ríos et al. (2010) identified a GCC transcription factor that belongs to a subgroup of the GARP (GOLDEN2, ARR-B, and Psr1) subfamily of MYB-related proteins containing a coiled-coil domain, designated GCC (GARP and coiled-coil), in *Citrus clementina* through differential transcriptomic analyses (a *Citrus* microarray containing approximately 20,000 cDNA fragments) associated with fruit color changes in two mutants (*39B3* and *39E7*) differing in delayed color break (carotenoid accumulation and chlorophyll degradation), and they observed that *CcGCC1* expression was strongly induced at the onset of color change in the flavedo and that the expression of *CcGCC1* was associated with peel ripening [16]. Gómez-Gómez et al. (2012) isolated a putative *CsMYB1* transcription factor gene from *Crocus* stigmas (saffron) by comparing the expression patterns during development in *C. sativus* (from yellow to the dark red stage) and two species with branched stigmas and yellow color (*C*. *cancellatus* and *C*. *pulchellus*), and they observed that the *CsMYB1* transcription factor gene was highly expressed in *C. sativus* stigmas in the dark red stage [17]. Grassi et al. (2013) generated the expression patterns of 19,324 genes from a transcriptomic analysis of ripening watermelon (*Citrus lanatus*) fruits by Illumina RNA sequencing, and RNA-Seq expression profiling yielded a set of transcription factor genes showing a positive correlation with ripening and carotenoid accumulation. Among them, 19 were homologous to tomato carotenoid-related genes (2 *RIN-MADS* (MADS-box transcription factor; 2 *TAGL1* (TAGL1 transcription factor); 5 *CNR* (COLORLESS NON RIPENING squamosa promoter binding-like protein); 1 *NAC-NOR* (NAC domain protein); 3 *SIAP2a* (AP2 transcription factor); 2 *SlERF6* (ERF6 mRNA), 1 *DET1* (*deetiolated1* homolog), 1 *DDB1* (UV-damaged DNA binding protein 1); 1 *CUL4* (cullin 4); and 1 *HB-1* (homeodomain leucine zipper protein)), all homologous to *S. lycopersicum*. Among these, six (three *CNR*, one *SIAP2a*, one *SlERF*6, and one *HB-1*) were differentially expressed in the flesh during fruit development and ripening [18]. Ye et al. (2015) performed a coexpression analysis of transcription factors and biosynthetic genes of the ascorbic acid, carotenoid, and flavonoid biosynthesis pathways to identify putative transcription factors that might regulate these biosynthetic processes; for this goal, the transcriptomes of tomato ‘Ailsa Craig’ and ‘HG6-61′ at seven developmental stages were evaluated using the Illumina sequencing platform, and the coexpression analysis revealed transcription factor genes with expression profiles that correlated with the patterns of genes related to each biosynthetic pathway during fruit development and ripening. In the case of carotenoid biosynthesis, 37 transcription factor genes, including a *MADS-box* gene and 4 *CCT* domain transcription factors, showed positive coexpression with carotenoid-biosynthesis-related genes. Other transcription factors, such as the *AUX/IAA*, *bHLH*, *MYB*, *SBP-box*, transcription factor *B3,* and *zinc finger*
*protein* genes, also exhibited a high correlation with carotenoid metabolism, suggesting a complex underlying regulatory network [19]. Jiang et al. (2019) used the RNA-Seq technique to investigate candidate genes of carotenoid metabolism in the flesh of pummelo (*Citrus maxima*) ‘GXMY’ and its mutants ‘HRMY’ and ‘HJMY’ in three fruit developmental stages and found 12 differentially expressed genes related to carotenoid biosynthesis; a coexpression analysis revealed that the expression patterns of several transcription factors, such as *bHLH*, *MYB*, *ERF*, *NAC,* and *WRKY**,* were highly correlated with differentially expressed genes involved in carotenoid biosynthesis [20]. Lu et al. (2019) identified six transcription factors (*CmMYB305*, *CmMYB29*, *CmRAD3*, *CmbZIP61*, *CmAGL24*, and *CmNAC1*) in *Chrysanthemum* X *morifolium* (a pink chrysanthemum cultivar, ‘Jianliuxiang Pink’, and its three bud sport mutants (including the white, yellow, and red color mutants ‘Jianliuxiang White’, ‘Jianliuxiang Yellow’, and ‘Jianliuxiang Red’, respectively)), using a weighted gene coexpression correlation network analysis (WGCNA) combined with a correlative analysis to determine whether they played an important role in carotenoid accumulation by regulating structural genes related to the carotenoid metabolism pathway and plastid development [21].

In the present work, we applied a coexpression analysis method developed by our research team [22] to identify transcription factor candidate genes possibly involved in the regulation of each carotenoid-biosynthesis-related structural gene in chili pepper fruit transcriptomes at different developmental stages from domesticated and wild accessions. This approach was complemented with the confirmation of cis-sequences in the promoters of the carotenoid-biosynthesis-related structural genes for each potentially selected transcription factor as a second criterion of selection.

## 2. Results

### 2.1. Expression Profiles of Carotenoid-Biosynthesis-Related Genes

Transcriptomic data were obtained from 12 accessions of *Capsicum annuum*: 4 wild, 6 domesticated, and 2 reciprocal crosses between a domesticated and a wild-type accession (Table 1). Fruits were collected at seven time points (0, 10, 20, 30, 40, 50, and 60 days after anthesis (DAA)) during the growth, development, and ripening of chili peppers. Data from all RNA-Seq libraries were analyzed as standardized expression profiles (SEPs). SEPs are seven dimensional numeric vectors in which each one of the numbers summarizes the standardized mean expression (mean = 0 and standard deviation = 1) at the corresponding time point based on ternary models in scale-free measurements [22,23].

The carotenoid structural biosynthetic genes used as target genes (TGs) (Figure 1) for coexpression analysis are summarized in Table 2. One SEP of *PDS*, *Z-ISO*, *LCYE*, *CYP97A*, *VDE*, *ZEP,* and *CCS* was identified in *Salsa*, but for *ZDS*, *CRTISO,* and *LCYB*, two different transcription profiles were identified for each gene, and both SEPs were analyzed for TF candidate prediction. Moreover, three transcript types encoding the *PSY* gene, and two types of *BCH* were identified in the *Salsa* database (Table 2). Nevertheless, only *PSY1* and *BCH1* presented a similar expression pattern as previously reported [24]. The expression profiles of carotenoid structural genes were grouped according to their pattern similarity.

The averages of the gene expression profiles of 12 accessions of *Capsicum annuum,* instead of individual SEPs, are presented in Figure 2. *PSY1*, *PDS,* and *ZDS2* showed transcript accumulation in the flower (0 DAA), although the transcripts of the *PSY* and *PDS* genes decreased during the growth of chili pepper fruit, while the transcripts of *ZDS2* exhibited a slight increase at 20 DAA and then decreased. At close to 50 DAA, a notable increase in the expression levels was observed for these three genes. In addition, the *BCH1* and *CCS* genes showed little or no accumulation of transcripts in the flower and growing fruit (10–40 DAA), although the expression increased at 50 and 60 DAA. *PSY1*, *BCH1,* and *CCS* reached their highest levels of expression at 60 DAA, whereas the transcript levels of *PDS* and *ZDS2* at 60 DAA were only slightly higher than at 50 DAA (Figure 2a). A similar pattern of expression was observed for the *ZDS1* gene, with an accumulation of transcripts in flowers, an increase at 40 DAA, a subsequent slight decrease at 50 DAA, and then maximum levels reached at 60 DAA (Figure 2a).

The *Z-ISO* and *LCYB1* genes displayed a low accumulation of transcripts at 0 DAA and a slight decrease at 10 DAA; nevertheless, a slight increase in the levels of expression was detected from 10 to 40 DAA, followed by a drastic increase at 50 DAA and a subsequent decrease at 60 DAA (Figure 2b).

In contrast, *CRTISO1, LCYE, LCYB2,* and *ZEP* reached their maximum levels of expression at 0 DAA (flower), with an abrupt decrease in the accumulation of transcripts at 10 DAA (Figure 2c). The expression levels of *LCYE* and *ZEP* remained low from 10 to 40 DAA, with a slight decrease at 50 and 60 DAA. Transcripts of *CRTISO1* gradually increased from 30 to 60 DAA but never reached the levels of expression detected in flowers. Similarly, an increase in the accumulation of *LCYB2* transcripts at 20 DAA was observed, which then decreased at 50 and 60 DAA (Figure 2c).

Finally, different types of transcription profiles were observed for *CRTISO2* and *CYP97A.* In the case of *CRTISO2,* a high accumulation of transcripts was detected from 0 to 40 DAA, and a subsequent decrease was observed at 50 and 60 DAA. The expression profile of *CYP97A* showed a sharp increase from 0 to 10 DAA, when the highest level of transcripts was reached, followed by an abrupt decrease at 20 DAA and then a slow decrease from 30 to 60 DAA (Figure 2d).

### 2.2. Coexpression Analysis of Carotenoid-Biosynthesis-Related Genes and TF Genes

The prediction of transcription factor (TF) gene candidates as possible participants in the regulation of carotenoid-biosynthesis-related genes was carried out by a coexpression analysis using the function g2g.TFcandidates, which is available in the R package *Salsa* v 1.0 [25]. Hypothesizing that TGs and TF genes should show highly similar expression profiles, strong and significant positive correlations between the SEPs of TFs and TGs were expected and evaluated. The first search for TF gene candidates was performed using the minimum r^2^ = 0.7, implying that all relations should have r^2^ > 0.7 to be considered “winners” and should appear in the output. Robustness was evaluated by the number of times that a correlation between the expression profiles of TG and TF genes was independently detected in different accessions at the fixed false discovery rate (FDR) selected by the researcher, and a minimum number of accessions (n.min.acc) = 12. For all analyzed TGs, the list of TF “winners” and the statistical data of coexpression analyses are compiled in Appendix A.

As a second criterion for the selection of TF candidates, we searched for putative binding sites for TF “winners” in the regulatory regions of TGs. The localization of TGs in the *Capsicum* reference genome, the prediction of consensus sequences of the promoter region (the TATA box, initiator element (INR), and downstream promoter element (DPE)), and the putative transcription start site (TSS) are shown in Appendix A. On the other side, TF “winners” with putative binding sites located in TG promoters and selected as good candidates to regulate the expression of structural genes of carotenoid biosynthesis and their *Arabidopsis* locus names and *S. lycopersicum* orthologs are included in Appendix A. Finally, Figure 3 shows the average SEPs of each analyzed TG and the four best candidates selected in each case.

As a result of the analysis of RNA-Seq, transcripts of 1859 transcription factors were identified during the growth and ripening process of *Capsicum* spp. After the coexpression analysis, we selected 83 TFs as possible regulators of the different carotenogenic structural genes (Appendix A). From them, putative binding sites in the promoters of those structural genes were identified for only 54 TFs (Appendix A).

By applying stringent parameters (min.fdr = 0.1, min.r^2^ = 0.6, and n.min.acc = 12), we obtained six good candidates for the transcriptional regulation of the *PSY1* gene (Appendix A). All of them presented a mean value for r (r.Med) < 0.94; nevertheless, the analysis of the promoter region of the *PSY* gene showed eight putative motif union sites just for PREDICTED NAC domain-containing protein 72-like (NAC72) and one putative binding site for Homeobox-leucine zipper protein ATHB-12-like (HDZIP12), (Figure 3a, Appendix A).

In the case of *PDS*, a coexpression analysis using stringent parameters did not identify appropriate TFs for its regulation; therefore, additional stringent parameters were tested (min.fdr = 0.1, min.r^2^ = 0.6, and n.min.acc = 8). The algorithm displayed four TF candidate genes with similar expression profiles as *PDS*: NAC domain-containing protein 83 (NAC83), a putative WRKY transcription factor 31 (WRKY31), NAC domain-containing protein 1 (NAC1), and WRKY transcription factor 71 (WRKY71) (Appendix A). Putative binding sites were identified for all these TFs (Figure 3b, Appendix A); accordingly, these TFs might be good candidates for *PDS* regulation for only 8 of the 12 analyzed accessions (Appendix A).

Less strict parameters were used to search for TF gene candidates possibly involved in the transcriptional regulation of *Z-ISO*: min.fdr = 0.1, min.r^2^ = 0.6, and n.min.acc = 6. The transcriptional profile of three TF genes showed a positive correlation (r.Med < 0.87) with the *Z-ISO* expression profile in seven accessions of *Capsicum annuum* (Appendix A). A regulatory region analysis in the *Z-ISO* promoter showed 12 and 11 putative binding motifs for WRKY transcription factor 7 (WRKY7) and WRKY71, respectively (Appendix A), and both were considered good candidates to regulate the expression of this carotenogenic gene (Figure 3c).

As mentioned above, two different expression profiles for *ZDS* were identified in the *Salsa* database with the same protein Id (*ZDS1* and *ZDS 2*) (Table 2 and Figure 2a); thus, a coexpression analysis was carried out independently for both. Using min.fdr = 0.1, min.r^2^ = 0.6, and n.min.acc = 6 as parameters, five TF candidate genes with a positive correlation (r.Med > 0.85) with that of *ZDS1* were predicted by g2gTF.candidates (Appendix A). Among them, an analysis of the *ZDS* promoter showed binding sites only for a putative WRKY transcription factor 29 (WRKY29) and a transcription factor MYBS3 (MYBS3) (Figure 3d and Appendix A); therefore, both were selected as candidates to regulate the transcription of this gene in seven and six accessions, respectively (Appendix A). By using the same search parameters, the *Salsa* algorithm selected two TFs with a positive correlation (r.Med > 0.85) with *ZDS2*: WRKY transcription factor 22 (WRKY22) and Dof zinc finger protein DOF3.1 (DOF3.1) (Figure 3e and Appendix A). For these TFs, 3 and 36 putative binding sites in the *ZDS* promoter were identified, respectively (Appendix A). Both TF candidate genes showed a positive correlation with the *ZDS* expression profile in six accessions; however, although *WRKY22* was not selected in CW and the *DOF3.1* was not identified in ZU, both were ultimately considered as good TF candidates for the six mentioned accessions of *Capsicum annuum* (Appendix A).

In the same way, two distinct SEPs with contrasting expression patterns for *CRTISO* were identified in *Salsa* (*CRTISO1* and *CRTISO2*) (Table 2, Figure 2c,d). They showed different patterns of expression during the growth and ripening processes of *Capsicum* fruits, and both were subjected to coexpression testing. A correlation analysis of *CRTISO1* was performed using min.fdr = 0.05, min.r^2^ = 0.7, and n.min.acc = 6 as selection parameters. Approximately 80 candidates with a positive correlation were displayed by the algorithm. The five best options (r.Med > 0.9) and the number of putative binding sites in the *CRTISO1* promoter are summarized in Appendix A. Moreover, a coexpression analysis of *CRTISO2* using less strict parameters (min.fdr = 0.1, min.r^2^ = 0.6, and n.min.acc = 7) showed seven candidates with r.Med ≥ 0.85 (Appendix A). The *CRTISO2* promoter analysis showed binding motifs for five of them: transcription factor BIM2-like isoform X4 (BIM2), auxin response factor 3 (ARF3), transcription factor MYB1R1 (MYB1R1), transcription factor TGA7 (bZIP_TGA7), and a putative WRKY transcription factor 20 isoform X1 (WRKY20) (Appendix A). In the case of *CRTISO1*, *CRTISO2,* and other analyzed TGs, only the first four TF “winners” with putative binding sites in their promoter region were graphed in Figure 3f,g; data of the other selected TF candidates can be consulted in Appendix A.

A coexpression analysis for prediction of TF candidates to regulate *LCYE* rendered results in five accessions (CO, CQ, SR, ST, and SY) using less strict parameters: min.fdr = 0.1, min.r^2^ = 0.6, and n.min.acc = 5. For these accessions, the g2g.TFcandidates algorithm identified 11 options with a positive correlation (r.Med > 0.9) between TG and TF SEPs (Appendix A). However, the *LCYE* promoter analysis showed binding sites for only eight of them: PREDICTED transcription factor bHLH79 (bHLH79.2), Dof zinc finger protein DOF2.5-like isoform X2 (DOF2.5), zinc finger protein NUTCRACKER (C2H2), homeobox-leucine zipper protein ATHB-6 (HDZIP6), PREDICTED homeobox-leucine zipper protein HAT5 (HDZIP_HAT5.2), Dof zinc finger protein DOF2.1 (DOF2.1), Dof zinc finger protein DOF1.4 (DOF1.4.1), and Dof zinc finger protein DOF3.6 (DOF3.6) (Appendix A). Figure 3h presents the coexpression patterns of *LCYE* and the first four selected TF gene candidates.

Two SEPs with different expression patterns were identified for *LCYB* (*LCYB1* and *LCYB2*) (Table 2, Figure 2b,c). Therefore, both SEPs were subjected to a coexpression analysis using nonstringent parameters: min.fdr = 0.01, min.r^2^ = 0.6, and n.min.acc = 6 (Figure 3i,j). For *LCYB1,* three TF “winners” with r.Med > 0.8 were identified; however, the analysis of the regulatory region showed only cis sequences for the transcription factor SRM1 isoform X1 (MYB_SRM1) in only six accessions (Figure 3i and Appendix A). Moreover, the *Salsa* algorithm identified four SEPs that showed a positive correlation (r.Med > 0.85) with *LCYB2* SEPs (Appendix A). Among them, binding motifs were located only for PREDICTED homeobox-leucine zipper protein HAT5 (HDZIP_HAT5.2) (Appendix A); therefore, this TF represented a good candidate to regulate *LCYB2* in nine accessions of *Capsicum annuum*.

Using the parameters min.fdr = 0.1, min.r^2^ = 0.6, and n.min.acc= 12, four TF “winners” were suggested as candidates for *CYP97A* regulation by a coexpression analysis with a correlation coefficient of r.Med < 0.9 (Appendix A). Interestingly, the analysis of cis sequences in the promoter of *CYP97A* revealed up to 56 binding motifs for Dof zinc finger protein DOF1,4 (DOF1.4.2) and 13 putative binding motifs for trihelix transcription factor GTL1-like (THLX1) (Figure 3k and Appendix A).

On the other hand, very stringent parameters (min.fdr = 0.05, min.r^2^ = 0.7, and n.min.acc = 12) were applied to the coexpression analysis between *BCH1* and the TF gene candidates. Eight TF “winners” were displayed by the algorithm, and remarkably, at least six TFs presented a positive correlation of r.Med > 0.95 (Appendix A). A minimum of 2 and a maximum of 40 binding sites were localized in the *BCH* promoter region for six TFs: MADS-box protein 04g005320 (MADS04g), NAC domain-containing protein 2 (NAC2), trihelix transcription factor DF1 (THLX_DF1), PREDICTED NAC domain-containing protein 72-like (NAC72), TGACG-sequence-specific DNA-binding protein TGA-2.1 (bZIP_2.1), and homeobox-leucine zipper protein HAT5 (HDZIP_HAT5.3) (Appendix A). Although only the four candidates with the highest r.Med are shown in Figure 3l, the six selected TFs can be considered good candidates for *BCH* transcriptional regulation.

A positive correlation (r.Med ≥ 0.92) between the SEPs of 13 TFs and *ZEP* was identified based on a coexpression analysis using very strict parameters (min.fdr = 0.05, min.r^2^ = 0.7, and n.min.acc = 12) (Appendix A). An analysis of the *ZEP* promoter region allowed us to find putative binding sites for nine candidates proposed by the g2g.TFcandidates algorithm: MYB-related protein 2 isoform X1 (MYB2), PREDICTED transcription factor bHLH122-like (bHLH122), Dof zinc finger protein DOF1.4 (DOF1.4.1), MYB-like transcription factor 4 (MYB4), homeobox protein 13-like isoform X2 (HDZIP13), PREDICTED transcription factor HBI1 (bHLH_HBI1), Dof zinc finger protein DOF2.4-like (DOF2.4), transcription factor TCP4 (TCP4), and PREDICTED transcription factor DIVARICATA-like isoform X1 (MYB_DIV) (Appendix A), and the expression pattern was apparently similar in all TF SEPs (Figure 3m).

Less strict parameters (min.fdr = 0.1, min.r^2^ = 0.6, and n.min.acc = 8) were employed to search for TF candidates for *VDE.* Only two TFs with low correlation coefficients (r.Med > 0.8) were displayed by the *Salsa* algorithm: an ethylene-responsive transcription factor-like protein, At4g13040, and a chloroplastic zinc finger protein, VAR3 (Appendix A). Unfortunately, no putative binding sites for these TFs were localized in the *VDE* promoter region. For these reasons, these results are not shown.

A coexpression analysis was carried out between TF SEPs and *CCS* transcripts using very strict parameters (min.fdr = 0.05, min.r^2^ = 0.6, and n.min.acc = 12), and it showed at least six TFs with a positive correlation (r.Med ≥ 0.94) (Appendix A). The search for putative binding sites in the promoter of *CCS* detected several binding motifs for the TF candidates selected by the *Salsa* algorithm: MADS-box protein 04g005320 (MADS04g), ethylene-responsive transcription factor ERF113 (ERF113), trihelix transcription factor DF1 (THLX_DF1), NAC domain-containing protein 2-like (NAC2), and PREDICTED TGACG-sequence-specific DNA-binding protein TGA-2.1 (bZIP_2.1) (Appendix A). The expression patterns of *CCS* and the TF candidates with r.Med >95 are shown in Figure 3n.

## 3. Discussion

Carotenoids are important pigments in plants because they are involved in essential processes, such as photosynthesis, photoprotection, and photomorphogenesis, and they are also precursors for the biosynthesis of the growth regulators abscisic acid (ABA) and strigolactones [26]. The biosynthesis of carotenoids is regulated at different levels. In the first instance, it depends on the availability of isoprenoid substrates; moreover, they are points in the pathway that controls the flux of metabolites through the pathway [27]. Environmental conditions, such as light, are important regulators of the carotenogenic pathway because they stimulate the biosynthesis of carotenoids and lead plastid structures to accumulate pigments [28]. Additionally, the biosynthesis of phytoene represents a rate-limiting step of the carotenoid biosynthesis pathway; the differential expression of *PSY* is regulated by abiotic factors such as temperature, photoperiod, salt, drought, high light, and ABA [29]. Finally, at the genetic level, carotenoid biosynthesis is regulated at the transcriptional, post-transcriptional, and epigenetic levels [27].

In *Capsicum* spp., the content of carotenoids in fruits is variable and changes from green or yellow in immature stages to red, orange, and yellow in mature fruits [1]. It has been reported that the *PSY*, *LCYB*, *BCH*, and *CCS* genes are necessary for the synthesis of capsanthin, and variations in their expression patterns result in the variety of colors of chili pepper fruits [30]. In this sense, the accumulation of transcripts of some biosynthetic genes is congruent with the change in color in the fruits and the accumulation of carotenoids [24].

The transcriptional regulation of the carotenogenic pathway in chili pepper fruits is poorly understood, as mentioned previously. Few transcription factors have been identified as regulators of the expression of the genes encoding the carotenoid biosynthetic enzymes. In our present study, we identified TF candidates that might be involved in the transcriptional regulation of these carotenogenic enzymes by using coexpression analysis and searching for putative binding sites for each TF candidate in the promoter of structural genes of the carotenoid biosynthesis pathway. We consider that this approach represents a good option for selecting TF candidates for subsequent studies of function and characterization through different experimental approaches. In the present work, an analysis of transcriptomic data identified 1859 TFs that are expressed during the growth, development, and ripening of *Capsicum* spp. fruits. After the coexpression analyses, we obtained a list of 83 TF “winners”. After the analysis of the binding sites of these TFs in the promoter regions of the carotenoid-biosynthesis-related structural genes, it was possible to compile a list of 54 TF candidates susceptible to experimental assessment. A correlation analysis was previously used with a similar purpose in *Capsicum* spp. A positive correlation between the expression patterns of *CaERF82*, *CaERF97*, *CaERF66*, *CaERF107,* and *CaERF101* and the accumulation patterns of carotene, zeaxanthin, and capsorubin suggested their possible participation in the transcriptional regulation of carotenoid biosynthesis [5]. Additionally, the carotenoid biosynthesis profile of lutein, zeaxanthin, and capsorubin in the pericarp showed a positive correlation with the expression profile of the *CabHLH009*, *CabHLH032*, *CabHLH048*, *CabHLH095,* and *CabHLH100* genes [31]. A coexpression analysis was also used to identify TF candidates for the regulation of the carotenoid biosynthesis pathway in *Capsicum* spp. A positive correlation between the expression pattern of the *CCS* gene and the TF genes *CaDIV1* and *CaMYB3R-5* was previously reported [6]. Finally, by using a coexpression analysis, U-box domain-containing protein 52, GATA transcription factor 26, RING/FYVE/PHD-type, F-box protein SKIP23, CONSTANS-LIKE 9, and zinc finger family FYVE/PHD-type were suggested as possible transcriptional regulators of *CCS* in *Capsicum* [7]. Interestingly, our coexpression approach combined with an analysis of the promoter regions of structural carotenoid biosynthetic genes did not recover any of the candidates mentioned above (Appendix A).

The described functions for the families of the TF candidates compiled in Appendix A are associated with the regulation of responses to different biotic and abiotic factors, such as pathogens, cold, drought, and salt stress, among others. This was also the case for the TF CaMADS, which was found to play an important role in the responses to cold, salt, and osmotic stress in chili pepper [32]; in our analysis, only one TF candidate belonging to the MADS family (MADS04g) was selected by the g2g.TFcandidates algorithm, and it was proposed as a candidate to regulate the transcription of *BCH* and *CCS.* Interestingly, 9 and up to 56 putative binding sites for this TF were identified in the promoters of *BCH* and *CCS*, respectively (Appendix A). MADS04g (previously reported as CaMADS and CaMADS-RIN) is an important TF associated with fruit development/ripening in chili pepper [33,34,35]. In addition, its ortholog in tomato, SlCBM1 (MADS-box protein 04g005320, NCBI reference sequence: NP_001362848.1) was reported to be a positive regulator of tomato fruit ripening, in particular a regulator of ethylene biosynthesis and a transcriptional regulator of *PSY*, *PDS*, *LCYB,* and *LCYE* during carotenoid accumulation [36,37]. Thus, it is plausible to hypothesize that some TFs described as regulators of stress responses could also have a role during the growth, development, and ripening of fruits. Several functions have been assigned to MYB TFs in *Capsicum* spp. fruits; for instance, they have been reported to be involved in the accumulation of anthocyanins [38] and in the regulation of capsaicinoid biosynthesis [39,40] and cold stress [41]. In this work, members of the MYB family were selected as putative regulator candidates of carotenoid biosynthetic genes (Figure 3d,g,i,m; Appendix A); previously, Arce-Rodríguez et al. (2021), using a coexpression analysis, proposed *CaDIV1* (XP_016551180.1) as a candidate to regulate *CCS* only in the ST accession, which corresponds to MYB_SRM1 (XP_016551180.1), which was suggested as a regulator of *LCYB1* in the present study (Figure 3I) in six accessions of *Capsicum annuum* (Appendix A). Recently, a dual function of an MYB TF was described: the knockdown of *CaMYB306* in chili pepper resulted in decreased expression of carotenogenic structural genes in fruits. Additionally, the overexpression of this gene in *S. lycopersicum* plants increased the accumulation of carotenoids and showed earlier coloration of fruits. On the other hand, CaMYB306 was described as a negative regulator of cold resistance in chili pepper [8]. These results suggest that TFs that have been characterized as regulators of stress responses could also be involved in the regulation of carotenoid biosynthesis.

Two members of the Trihelix TF family (THLX1 and THLX_DF1) were identified as candidates for *CYP97A*, *BCH,* and *CCS* regulation. THLX_DF1 belongs to the GT-2 group and is involved in responses to pathogens; salt stress; flower, trichome, and stoma development; seed abscission; and embryogenesis. Moreover, the orthologs of THLX1 and THLX_DF1 in tomato (*S. lycopersicum*; *SlGT*-30) were demonstrated to be ubiquitously expressed in buds, flowers, leaves, and fruits [42]. Interestingly, Trihelix DF1 has been reported to bind to promoters of genes that are downregulated by light and dark-inducible [43,44].

At least six TFs belonging to the WRKY TF family (WRKY7, WRKY20, WRKY22, WRKY29, WRKY31, and WRKY71) were identified as candidates to regulate some points in the first steps of the carotenogenic pathway in this work. These TFs have been typically associated with environmental stress responses and defense, although important levels of the transcripts of several *CaWRKYs* have been detected during fruit ripening [45]. For example, recent evidence has demonstrated that WRKY9 could be associated with pungency and capsaicinoid content in chili pepper fruits [46]. Among the TF candidates proposed in this work, CaWRKY22 has been described as a positive regulator of innate immunity against *Ralstonia solanacearum.* This process is regulated by a signaling pathway mediated by jasmonic acid, salicylic acid, and ethylene and involves a network of WRKY TFs [47].

Regarding the NAC TF family, we propose four candidates as regulators of carotenoid biosynthesis: NAC1, NAC2, NAC72, and NAC83. Previous reports of NAC TFs in *Capsicum* spp. have associated them with biotic and abiotic stress; for instance, CaNAC1 was found to be induced during incompatible interactions between chili pepper and pathogens (viruses and bacteria) [48], CaNAC2 was found to be involved in the response to abiotic stress tolerance [49], CaNAC064 was found to be associated with cold stress tolerance [50], and CaNAC1 was proposed to regulate low-temperature-induced phospholipid degradation in green bell pepper [51]. In the same way, the orthologs of these four NAC candidates in *S. lycopersicum* have been grouped phylogenetically in the same cluster (Va) and associated with stress responses [52]. However, in tomato, NAC TFs have been associated with fruit ripening and the accumulation of carotenoids, particularly *SlNAC1* and *SlNAC4* (ortholog of *NAC2* selected as candidates in this work) [53,54]. Moreover, NAC TF (NOR-like1) has been identified as a positive regulator of fruit ripening because it is involved in the formation of color [55]. In particular, SlNAC4 is involved in the transcriptional regulation of the *PSY*, *LCYB,* and *LCYE* structural genes of the carotenoid biosynthesis pathway [37].

Additionally, seven different transcription factors of the DOF family were identified as TF candidates in the present study (Appendix A), and they have been associated with seed development, dormancy, germination, and light-mediated processes [56,57].

Six candidates belonging to the HD-ZIP TF family were selected as candidates to regulate the biosynthesis of carotenoids in *C. annuum*. HDZIP (CaCBF) was reported as a cold-inducible TF [58], and CaHDZ27 was found to be a positive regulator of resistance to *R. solanacearum* [59]. In tomato, HD-ZIP homeobox genes have been associated with the control of ethylene, the regulation of floral organogenesis, and early fruit development [60]. In the present study, we propose two members of the bZIP TF family (bZIP_TGA7 and bZIP_TGA2.1) as TF candidates to regulate *CRTISO2* and *CCS*, respectively. In chili pepper, bZIPs have been involved in the modulation of ABA signaling and drought tolerance [61] and in environmental stress responses [62], while CabZIP2 has been proposed to participate in bacterial disease resistance [63,64].

Regarding the coexpression analyses in the accessions of *Capsicum* spp., these could only be performed for *PSY*, *BCH*, *CYP97A, CCS*, and *ZEP* in the 12 accessions, and they were not affected by the late ripening time of AS, CW, and JE, which mature at 80, 70, and 70 DAA, respectively [22]. A correlation analysis was carried out with *PDS, Z-ISO, ZDS, CRTISO, LCYE, LCYB*, and *VDE* in a minimum of five and a maximum of nine accessions; in these cases, the selected TFs predicted to be good candidates by the correlation analysis only apply for those accessions that were highly correlated.

## 4. Materials and Methods

### 4.1. Biological Materials, Sequencing, and RNA-Seq Data Analysis

Transcriptomic data were obtained from 12 accessions of *Capsicum annuum*: 4 wild, 6 domesticated, and 2 reciprocal crosses between domesticated and wild-type accessions (Table 1). Fruits were collected at seven time points (0, 10, 20, 30, 40, 50, and 60 days after anthesis (DAA)) during the growth and ripening process of chili pepper. The procedures of seed germination and growth, RNA-Seq library generation, and processing were as previously described [22]. A total of 84 sampling points (12 accessions x 7 time points) and two RNA-Seq libraries (biological replicates), 84 × 2 = 168 RNA-Seq libraries, were analyzed for time expression profile estimation. Data from 179 RNA-Seq libraries were obtained, and more than 3 billion raw reads were mapped to the reference chili pepper genome. These data were deposited in the NCBI gene expression omnibus (GEO) 25 and are accessible through GEO Series accession number GSE165448 (https://www.ncbi.nlm.nih.gov/geo/query/acc.cgi?Acc=GSE16 5448, accessed on 30 August 2022).

Data from all RNA-seq libraries were analyzed as standardized expression profiles (SEPs). SEPs are seven dimensional numeric vectors in which each of one of the numbers summarizes the standardized mean expression at the corresponding time point (0, 10, 20, 30, 40, 50, and 60 DAA) obtained from RNA-Seq in the 12 accessions of *Capsicum annuum*, based on ternary models in scale-free measurements [22,23].

### 4.2. Selection of TF Candidates

The prediction of transcription factor (TF) gene candidates as possible participants in the regulation of the carotenoid-biosynthesis-related genes was carried out by a coexpression analysis using the function g2g.TFcandidates, which is available in the R package *Salsa* v 1.0. [25]. Hypothesizing that target genes (TGs) and TF genes should show highly similar expression profiles, strong and significant positive correlations between the standardized expression profiles (SEPs) of TFs and TGs were expected and evaluated. This method implements a search for TF gene candidates by evaluating Pearson’s correlations (r^2^) between the expression profile of the TG and the expression profiles of each one on the TF genes annotated in the *Capsicum* reference genome (also recorded in the transcriptome *Salsa* database) in turn with each of the accessions that we wanted to consider preferentially with the 12 accessions of *Capsicum*
*annuum*. By evaluating pairwise correlations between the TG and each TF gene profile in each accession at all stages of growth, development, and ripening, the algorithm filtered the “winners” (potential candidates using the set of thresholds input). The first search of TF gene candidates was performed using the minimum r^2^ for the correlations, which was set to min.r^2^ = 0.7, implying that all relations should have r^2^ > 0.7 to be considered “winners” and should appear in the output. Note that this threshold was equivalent to asking that ˆr > √0.7 = 0.84. Robustness was evaluated by the number of times that a correlation between the expression profiles of the TG and TF genes was independently detected in different accessions at the fixed false discovery rate (FDR) selected by the researcher. Thus, the stringent parameters employed for predicting TF gene candidates were min.fdr = 0.05, min.r^2^ = 0.7, and n.min.acc = 12. If the algorithm did not display options using these strict parameters, less stringent values were tested (min.fdr = 0.1, min.r^2^ = 0.6, and n.min.acc = 12). Finally, the number of accessions in which we wanted to identify TF candidates could be modified and tested once more. The statistical analyses were complemented by the maximum of the absolute differences (m.a) between the SEPs of the TG and TF genes. Values of m.a.Max > 1.5 were considered to contain extreme outliers. An m.a.Max < 0.5 was acceptable. Good TF gene candidates should have a “large” value of r^2^ and a “small” value of m.a. Complete statistical values of each of the TF “winners” are compiled in Appendix A. Finally, the TGs used for the coexpression analyses are summarized in Figure 2 and Table 2.

For the list of candidates displayed by g2g.TFcandidates, we analyzed the promoter region of each biosynthetic target gene to identify putative binding sequences for each TF candidate. Consensus DNA-binding sequences, including the TATA box, initiator element (INR), downstream promoter element (DPE), and putative transcriptional starting site (TSS) of each promoter region, were located using the YAPP Eukaryotic Core Promoter program (http://www.bioinformatics.org/yapp/cgi-bin/yapp.cgi, accessed on 24 May 2022), and the results of the promoter region location of each gene are summarized in Appendix A. The bp of the TSS column indicated the bp number from ATG. Then, 2500 bp from the TSS was used for searching cis-sequences. Putative binding sites for TF candidates were determined using PlantPan software and *Arabidopsis thaliana* as references [65]. The UniProt [66] and Plant Transcription Factors databases were used for the identification of orthologous genes of *Capsicum* spp. in *A. thaliana* [67]. Only the transcription factor candidates with putative binding sites located in the promoters of target genes are summarized in Appendix A.

## 5. Conclusions

In this work, we proposed a list of putative TF candidates associated with the regulation of the carotenoid biosynthesis pathway by RNA-Seq coexpression analysis, and further verification assays (ChIP, yeast one-hybrid, and virus-induced gene silencing, among others) are necessary to confirm or discard the participation of them in the transcriptional regulation of carotenoid biosynthesis. Due to the limited information about the transcriptional regulation of the carotenoid biosynthesis pathway in chili pepper and other nonclimacteric fruits, we propose a list of putative candidates that could guide future experimental approaches.

Some TF candidates, such as MADS04g and NAC2, have previously been reported as regulators of the ripening process in chili pepper and tomato, respectively. Thus, our RNA-Seq coexpression analysis represents a valuable tool for the analysis and selection of novel candidates to develop further functional TF gene studies and gene characterization. Gene function analysis using virus-induced gene silencing and the yeast one-hybrid assay of some of the selected candidates (ERF113, HDZIP_HAT5.3, MADS04g, NAC2, NAC72, and THLX_DF1) are currently underway.

It is important to mention that the TF “winners” listed in Appendix A for which it was not possible to locate cis-binding sequences in the promoters of carotenogenic genes cannot be ruled out as transcriptional regulators of the pathway, but they could act as regulators of other transcription factors and are susceptible to being evaluated by yeast two-hybrid assays.

Although most TFs of plants have been associated with the regulation of different processes other than the biosynthesis of carotenoids, there are several cases (addressed in the discussion section) where different members of families of TFs have been found to act as regulators of this biosynthetic pathway.

## Figures and Tables

**Figure 1 ijms-23-11774-f001:**
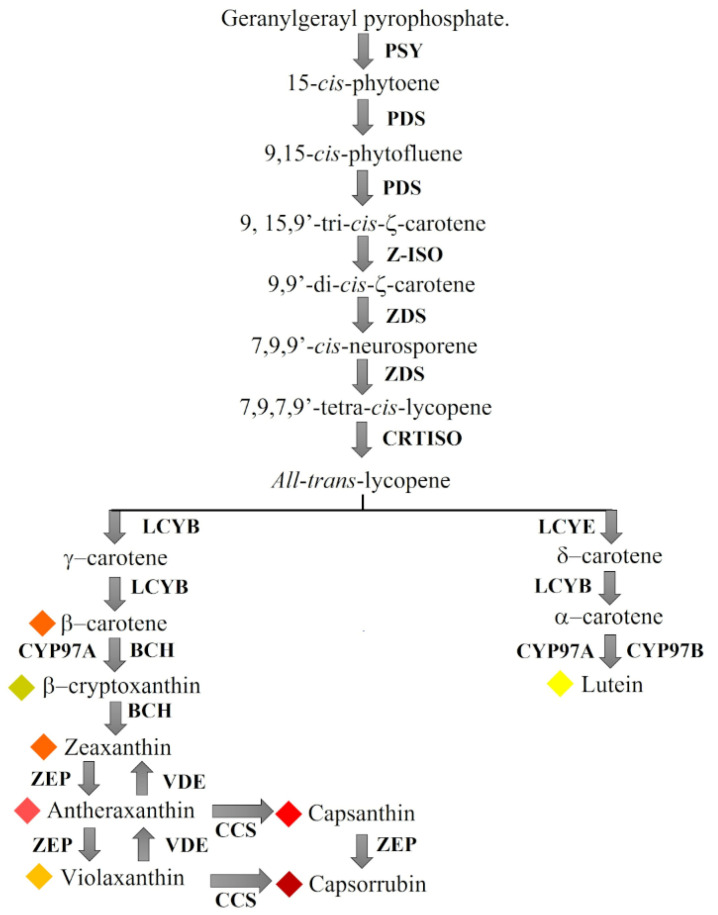
Chili pepper fruit carotenoid biosynthetic pathway. PSY (phytoene synthase), PDS (phytoene desaturase), Z-ISO (ζ-carotene isomerase), ZDS (ζ-carotene desaturase), CRTISO (carotene isomerase), LCYB (β-lycopene cyclase), LCYE (ε-lycopene cyclase), BCH (β-carotene hydroxylase), CYP (β-carotene hydroxylase cytochrome 450 types A and B), ZEP (zeaxanthin epoxidase), VDE (violaxanthin epoxidase), and CCS (capsanthin-capsorubin synthase). Modified from [1]. Colored rhombuses represent the color conferred by each carotenoid.

**Figure 2 ijms-23-11774-f002:**
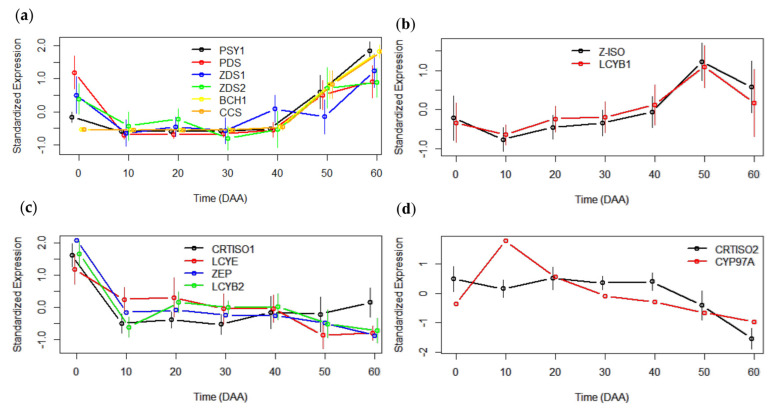
Groups of TGs of carotenoid biosynthesis with similar expression patterns. (**a**) Phytoene synthase (*PSY1*), phytoene desaturase (*PDS*), Z-carotene desaturase 1 (*ZDS1*), Z-carotene desaturase 2 (*ZDS2*), beta-carotene hydroxylase (*BCH1*), and capsanthin/capsorubin synthase (*CCS*) (**b**) Z-carotene isomerase (*Z-ISO*) and lycopene beta cyclase 1 (*LCYB1*) (**c**) lycopene isomerase 1 (*CTRTISO1*), lycopene epsilon cyclase (*LCYE*), zeaxanthin epoxidase (*ZEP*), and lycopene beta cyclase 2 (*LCYB2*) (**d**) lycopene isomerase 2 (*CRTISO2*) and cytochrome P450 97A (*CYP97A*).

**Figure 3 ijms-23-11774-f003:**
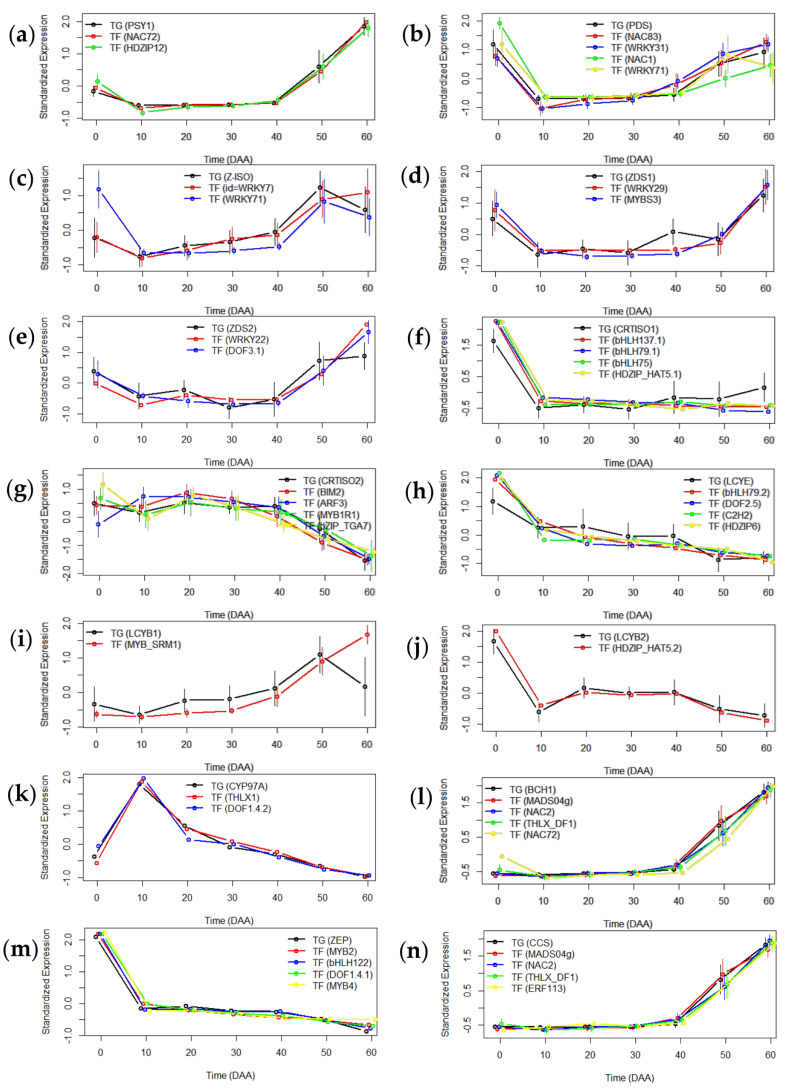
Average SEPs for TGs and the “TF winners”. (**a**) Phytoene synthase (*PSY1*), (**b**) phytoene desaturase (*PDS*), (**c**) Z-carotene isomerase (*Z-ISO*), (**d**) Z-carotene desaturase 1 (*ZDS1*), (**e**) Z-carotene desaturase 2 (*ZDS2*), (**f**) lycopene isomerase 1 (*CTRTISO1*), (**g**) lycopene isomerase 2 (*CRTISO2*), (**h**) lycopene epsilon cyclase (*LCYE*), (**i**) lycopene beta cyclase 1 (*LCYB1*), (**j**) lycopene beta cyclase 2 (*LCYB2*), (**k**) cytochrome P450 97A (*CYP97A*)*,* (**l**) beta-carotene hydroxylase (*BCH1*), (**m**) zeaxanthin epoxidase (*ZEP*), and (**n**) capsanthin/capsorubin synthase (*CCS*).

**Table 1 ijms-23-11774-t001:** Chili pepper accessions used for the RNA-Seq analyses.

Accession Type	Accession Name	Abbreviation
D	Ancho San Luis	AS
D	Criollo de Morelos 334 (CM334)	CM
D	California Wonder	CW
D	Jalapeño Espinalteco	JE
D	Serrano Tampiqueño 74	ST
D	Zunla-1	ZU
W	Piquín Coahuila	CO
W	Piquín Querétaro	QU
W	Piquín Sonora Red	SR
W	Piquín Sonora Yellow	SY
C	F1: CM female x QU male	CQ
C	F1: QU female x CM male	QC

D, domesticated; W, wild; C, crosses.

**Table 2 ijms-23-11774-t002:** Description of the analyzed target genes involved in carotenoid biosynthesis.

*Salsa* Id	Description	Abbreviation	Protein Id
20260	Bifunctional 15-*cis*-phytoene synthase, chromoplastic	*PSY1*	XP_016570422.2
10326	Phytoene synthase 2, chloroplastic	*PSY2*	XP_016560212.1
9117	Phytoene synthase 2, chloroplastic-like	*PSY3*	XP_016576229.2
31467	15-*cis*-phytoene desaturase, chloroplastic/chromoplastic	*PDS*	XP_016562405.2
22592	15-*cis*-zeta-carotene isomerase, chloroplastic	*Z-ISO*	XP_016550148.1
5981	Zeta-carotene desaturase, chloroplastic/chromoplastic	*ZDS1*	NP_001311497.1
5979	Zeta-carotene desaturase, chloroplastic/chromoplastic	*ZDS2*	NP_001311497.1
11712	Prolycopene isomerase, chloroplastic	*CRTISO1*	XP_016548432.1
28615	Prolycopene isomerase, chloroplastic	*CRTISO2*	XP_016555023.1
2529	Lycopene epsilon cyclase, chloroplastic isoform X2	*LCYE*	XP_016540361.1
4235	Lycopene beta cyclase, chloroplastic/chromoplastic	*LCYB1*	XP_016571836.1
27689	Lycopene beta cyclase, chloroplastic	*LCYB2*	XP_016543793.2
20137	Protein LUTEIN DEFICIENT 5, chloroplastic	*CYP97A*	XP_016551303.1
4572	Beta-carotene hydroxylase 1, chloroplastic	*BCH1*	NP_001311784.1
13055	Beta-carotene hydroxylase 2, chloroplastic isoform X1	*BCH2*	NP_001385279.1
28396	Zeaxanthin epoxidase, chloroplastic	*ZEP*	XP_016561102.1
16272	Violaxanthin de-epoxidase, chloroplastic isoform X1	*VDE*	XP_016550436.1
34360	Capsanthin/capsorubin synthase, chromoplastic	*CCS*	NP_001311998.1

Id, identifier.

## Data Availability

Data is available at the Gene Expression Omnibus (GEO) of the NCBI (https://www.ncbi.nlm.nih.gov/geo/query/acc.cgi?Acc=GSE16 5448, accessed on 30 August 2022).

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
