# Peer review of "Putative Transcription Factor Genes Associated with Regulation of Carotenoid Biosynthesis in Chili Pepper Fruits Revealed by RNA-Seq Coexpression Analysis"

_ijms, 2022, doi:10.3390/ijms231911774_

Round 1

Reviewer 1 Report

Comments to authors

Manuscript is very well written and interesting.

I have only minor comments for authors to address:

1) Line 192 Did you mean Table 2 instead Table 1?

2) Figure 2 maybe to put PSY1 instead PSY since you write in the manuscript text PSY1

3) Line 245 correct word coexpression.

Author Response

First of all, we would like to thank Reviewer 1 for all corrections/observations/comments to improve our manuscript. Here we present the point-by-point responses.

REVIEWER 1

Manuscript is very well written and interesting.

I have only minor comments for authors to address:

  • Line 192 Did you mean Table 2 instead Table 1?

Response:      Ok, it was corrected (line 192).

  • Figure 2 maybe to put PSY1 instead PSY since you write in the manuscript text PSY1.

Response: Figures 2 and 3 were modified according with your suggestion; we changed PSY by PSY1 and BCH by BCH1 in both cases.

3) Line 245 correct word coexpression.

Response: Ok, it was corrected (line 246).

Reviewer 2 Report

I get your article (Putative Transcription Factor Genes Associated with Regulation of Carotenoid Biosynthesis in Chili Pepper Fruits Revealed by RNA-Seq Coexpression Analysis) to review. Your article is very well organized and written as well as have parameters tested and your article. Although the carotenoid biosynthesis pathway in Capsicum has been extensively studied from different perspectives, the transcriptional regulation of the gene encoding the carotenoid biosynthetic enzyme in this fruit has not yet been elucidated. The interactions between transcription factors and promoters of carotenoid biosynthetic genes have been studied in several crops many times. Authors applied a co-expression analysis method developed by their research team to identify transcription factor candidate genes possibly involved in the regulation of each carotenoid biosynthesis-related structural gene in chili pepper fruit transcriptomes at different developmental stages from domesticated and wild accessions. There are a few small flaws here that you need to revise carefully.

1.     Please further confirm which transcription factor or factors are involved in the biosynthesis of carotenoids in pepper fruits.

2.     Whether the NAC1, NAC83, WRKY31 and WRKY71 transcription factors bind to the promoters of carotenoid synthesis genes can be verified by ChIP and yeast one-hybrid techniques.

3.     Please do further experiments to confirm which transcription factor interacts with which key gene for carotenoid synthesis to regulate carotenoid synthesis.

Once the above concerns are fully addressed, the manuscript could be accepted for publication in this journal.

Author Response

First of all, we would like to thank Reviewer 1 for all corrections/observations/comments to improve our manuscript. Here we present the point-by-point responses.

REVIEWER 2

I get your article (Putative Transcription Factor Genes Associated with Regulation of Carotenoid Biosynthesis in Chili Pepper Fruits Revealed by RNA-Seq Coexpression Analysis) to review. Your article is very well organized and written as well as have parameters tested and your article. Although the carotenoid biosynthesis pathway in Capsicum has been extensively studied from different perspectives, the transcriptional regulation of the gene encoding the carotenoid biosynthetic enzyme in this fruit has not yet been elucidated. The interactions between transcription factors and promoters of carotenoid biosynthetic genes have been studied in several crops many times. Authors applied a co-expression analysis method developed by their research team to identify transcription factor candidate genes possibly involved in the regulation of each carotenoid biosynthesis-related structural gene in chili pepper fruit transcriptomes at different developmental stages from domesticated and wild accessions. There are a few small flaws here that you need to revise carefully. 

1. Please further confirm which transcription factor or factors are involved in the biosynthesis of carotenoids in pepper fruits.

Response: There are no transcription factors involved in the biosynthesis of carotenoids confirmed in the literature. In the case of CaMADS (Dubey et. al., 2019), and CaMYB306 (Ma et. al., 2022), they were proposed as regulators of the ripening process and accumulation of pigments respectively in chili pepper, but as far as we know there were no confirmed results (ChIP or yeast one-hybrid assays) on what structural gene could be regulating. On the other hand, there are some candidates for the transcriptional regulation of CCS that were identified by coexpression analysis: GATA transcription factor 26, RING/FYVE/PHD-type, F-box protein SKIP23, CONSTANS-LIKE 9, and zinc finger family FYVE/PHD-type by Li et. al., 2021 and CaDIV1 proposed by Arce-Rodríguez et. al., (2020), but they have not yet confirmed experimentally.

2. Whether the NAC1, NAC83, WRKY31 and WRKY71 transcription factors bind to the promoters of carotenoid synthesis genes can be verified by ChIP and yeast one-hybrid techniques.

Response: Yes, those techniques were included (line 626), as part of the verification assays.

3. Please do further experiments to confirm which transcription factor interacts with which key gene for carotenoid synthesis to regulate carotenoid synthesis.

Response: Yes, we are currently carrying out gene function studies with some of the selected candidates (see page 18, lines 636-638 of the corrected manuscript version). We have at least 26 candidates selected in this work that must be further characterized regarding their function and the interactions with promoters of the carotenoid-related structural genes. This will be a very urgent task for us.

Once the above concerns are fully addressed, the manuscript could be accepted for publication in this journal.